# Evaluation of a Conformationally Constrained Indole Carboxamide as a Potential Efflux Pump Inhibitor in *Pseudomonas* *aeruginosa*

**DOI:** 10.3390/antibiotics11060716

**Published:** 2022-05-26

**Authors:** Yongzheng Zhang, Jesus D. Rosado-Lugo, Pratik Datta, Yangsheng Sun, Yanlu Cao, Anamika Banerjee, Yi Yuan, Ajit K. Parhi

**Affiliations:** TAXIS Pharmaceuticals, Inc., 9 Deer Park Drive, Suite J-15, Monmouth Junction, NJ 08852, USA; yzhang@taxispharma.com (Y.Z.); jrosado@taxispharma.com (J.D.R.-L.); pdatta@taxispharma.com (P.D.); yangsheng.sun@gmail.com (Y.S.); yanlucao@gmail.com (Y.C.); anamikabanerjee375@gmail.com (A.B.); yiyuan@taxispharma.com (Y.Y.)

**Keywords:** *Pseudomonas aeruginosa*, RND efflux pumps, antimicrobial drug resistance, efflux pump inhibitor, indole carboxamide

## Abstract

Efflux pumps in Gram-negative bacteria such as *Pseudomonas aeruginosa* provide intrinsic antimicrobial resistance by facilitating the extrusion of a wide range of antimicrobials. Approaches for combating efflux-mediated multidrug resistance involve, in part, developing indirect antimicrobial agents capable of inhibiting efflux, thus rescuing the activity of antimicrobials previously rendered inactive by efflux. Herein, TXA09155 is presented as a novel efflux pump inhibitor (EPI) formed by conformationally constraining our previously reported EPI TXA01182. TXA09155 demonstrates strong potentiation in combination with multiple antibiotics with efflux liabilities against wild-type and multidrug-resistant (MDR) *P.* *aeruginosa*. At 6.25 µg/mL, TXA09155, showed ≥8-fold potentiation of levofloxacin, moxifloxacin, doxycycline, minocycline, cefpirome, chloramphenicol, and cotrimoxazole. Several biophysical and genetic studies rule out membrane disruption and support efflux inhibition as the mechanism of action (MOA) of TXA09155. TXA09155 was determined to lower the frequency of resistance (FoR) to levofloxacin and enhance the killing kinetics of moxifloxacin. Most importantly, TXA09155 outperformed the levofloxacin-potentiation activity of EPIs TXA01182 and MC-04,124 against a CDC/FDA panel of MDR clinical isolates of *P. aeruginosa.* TXA09155 possesses favorable physiochemical and ADME properties that warrant its optimization and further development.

## 1. Introduction

Pseudomonas aeruginosa is an efficient opportunistic pathogen causing serious infections in patients who are mechanically ventilated, individuals who are immunocompromised, and patients with malignancies or HIV infection [1]. It is the most common Gram-negative organism identified in nosocomial pneumonia, and it can cause infection in immunocompromised hosts and infection in those with structural lung disease such as cystic fibrosis, surgical site infections, urinary tract infections and bacteremia [2]. In the Unites States, there are approximately 300,000 hospital-acquired pneumonia (HAP) infections per year, of which an estimated 64,800 are caused by *P. aeruginosa*, resulting in 27,200 annual deaths [3,4]. The pathogenicity of *P. aeruginosa* is largely caused by several virulence mechanisms such as secreted toxins, quorum sensing and biofilm formation, as well as genetic flexibility that enables it to survive in varied environments [1,2]. Lung injury associated with *P. aeruginosa* infection results from both the direct destructive effects of the organism on the lung parenchyma and exuberant host immune responses [1]. Treatment of *P. aeruginosa* presents a serious challenge since selection of the most appropriate antibiotic is complicated by the ability of *p. aeruginosa* to develop resistance to multiple classes of antibacterial agents, which can severely limit the therapeutic options for treatment of serious infections [5]. An even more serious therapeutic challenge is the prevalence of multidrug-resistant (MDR) *P. aeruginosa* strains prevalent among isolates from lower respiratory tract infections. Various means can allow *P. aeruginosa* to become MDR. One method involves importing multiple resistance mechanisms on mobile genetic elements containing β-lactamases, aminoglycoside phosphoryltransferase, aminoglycoside acetyltransferase, aminoglycoside nucleotidyltransferase and/or ribosomal methyltransferase, which impact β-lactam and aminoglycoside antibiotics. Another method is through mutational events leading to diminished permeability to antibiotics (i.e., loss of porins) or the overexpression of resistance mechanism, (i.e., AmpC cephalosporinase or efflux pumps) [5].

Efflux pumps in Gram-negative bacteria provide intrinsic antimicrobial resistance by facilitating the extrusion of a wide range of compounds [6,7,8]. Many of these efflux pumps in Gram-negative bacteria belong to the resistance-nodulation-cell division (RND) family of tripartite efflux pumps [9,10]. A good example of this is the RND efflux systems in *P. aeruginosa*, which consist of many identified efflux pumps that are largely responsible for its multidrug resistance [11]. These include four well-characterized multidrug efflux pump systems (MexAB-OprM, MexCD-OprJ, MexEF-OprN, and MexXY-OprM), which are prevalent in a significant percentage of clinical isolates of *P. aeruginosa*, and six RND efflux pumps (MexJK, MexGHI-OpmD, MexVW, MexPQ-OpmE, MexMN, and TriABC-OpmH) that might contribute to resistance at the clinic [12,13,14]. Approaches for circumventing efflux-mediated resistance require developing new antibiotics that are poor substrates of efflux pumps, or indirect antimicrobial agents capable of inhibiting efflux, thus rescuing the activity of antimicrobials previously rendered inactive by efflux. The lack of innovation in the development of new antibiotics makes efflux inhibition an attractive avenue to rejuvenate older antimicrobials to tackle the antibiotics resistance problems [14,15]. An ideal efflux pump inhibitor (EPI) would efficiently block these pumps, increasing the concentrations of the antimicrobials within the cell, thus rendering them effective again [16].

A variety of chemical scaffolds have been shown to act as EPIs [17,18,19,20]. However, the development of these compounds has stopped or stalled for various reasons [21,22,23,24,25]. Hence, the discovery of novel EPI scaffolds active against Gram-negative pathogens, particularly *P. aeruginosa*, with the potential of clinical use is still needed. TAXIS Pharmaceuticals is committed [26] to finding solutions to multidrug-resistant (MDR) bacterial infections and has previously published preliminary efforts on a diaminopentanamide class of potentiators [27,28] and more recently on a heterocyclic carboxamide class of potentiators that yielded TXA01182 [29]. TXA01182 potentiates monobactam, fluoroquinolones, sulfonamides, and tetracyclines against *P. aeruginosa*. Further, TXA01182 demonstrated a synergistic effect with levofloxacin against several MDR *P. aeruginosa* clinical isolates.

The present study aims to improve the EPI activity and overall pharmacological properties of TXA01182 in *P. aeruginosa*, particularly against MDR clinical isolates. This work produced TXA09155, a novel EPI that enhances the activity of multiple classes of antimicrobials with efflux liabilities in wild-type and MDR clinical isolates of *P. aeruginosa* while playing a minimal role in membrane disruption and avoiding the onset of resistance.

## 2. Results and Discussion

### 2.1. Identification and In-Depth Potentiation Evaluation of a Potent First-Generation Indole 2-Carboxamide EPI

Previously, TAXIS Pharmaceutical reported on the results of a synthetic screen seeking to replace the aryl alkyl head groups of a diaminopentanamide class of EPI with more druggable fragments in order to eliminate or diminish membrane disruption as a secondary MOA and improve metabolic and serum stability. This work produced TXA01182, a novel EPI that enhances the activity of multiple classes of antimicrobials with efflux liabilities in wild-type and MDR clinical isolates of *P. aeruginosa* while playing a minimal role in membrane disruption and avoiding the onset of resistance. In a hit-to-lead campaign for more powerful EPIs with better drug-like properties, the diamine side chain of TXA01182 was constrained at the C2-amine and the C4 positions to form a pyrrolidine containing EPI, TXA09155, with an identical indole head group and C2-stereochemistry (S) (Figure 1). The rationale for such a modification came from the well-established fact that conformation constraints and molecular flexibility have a strong effect on the activity of flexible molecules. There are many examples where conformationally constrained analogs are more bioactive or more specific than the non-constrained molecules [30,31,32]. In addition to the better potency, constraining flexible molecules may lead to drug candidates with better toxicity profiles and PK properties.

To show if such a modification increased the potencies of TXA09155, its potentiation effects on different antibiotics were compared with TXA01182. TXA09155 was first assayed for its antibacterial activity against *P. aeruginosa* ATCC 27853, and the compound had a MIC of 50 µg/mL. As presented in Table 1, at 6.25 µg/mL, TXA09155 reduces the MICs of all tested antibiotics two-fold better than TXA01182, thus validating our design rationale.

Once it was confirmed that constraining the diamine side chain of TXA01182 resulted in a more active EPI, an in-depth evaluation of TXA09155′s potentiation abilities on multiple antibiotics over a concentration range was carried out, and the results are presented in Table 2. The ability of TXA09155 to lower the MIC of multiple antibiotics with efflux liabilities in *P. aeruginosa* was tested at concentrations ranging from 25 to 3.13 µg/mL (1/2 to 1/16th MIC). TXA09155 potentiated most antibiotics in a concentration-dependent manner except for levofloxacin where all active concentrations showed the same level of potentiation. Fluoroquinolones levofloxacin and moxifloxacin were potentiated by up to 16- and 128-fold, respectively. Tetracyclines doxycycline and minocycline were potentiated by up to 32- and 256-fold, respectively. Cefpirome, a cephalosporin, was potentiated up to 16-fold. Aztreonam, a monobactam, was potentiated up to 32-fold. Chloramphenicol and cotrimoxazole were potentiated up to 128-fold. As anticipated for an EPI, imipenem, which is not the substrate of RND efflux pumps in *P. aeruginosa*, was not potentiated by TXA09155 at any of the concentrations tested. These results are in line with TXA09155, maintaining the EPI activity of its parent compound TXA01182. The lowest active concentration for TXA09155 was 6.25 µg/mL. Thus, the 6.25 µg/mL concentration was chosen for future antimicrobial potentiation assays in *P. aeruginosa* to ensure that its MIC did not interfere with its potentiation properties.

### 2.2. TXA09155 Plays a Minimal Role in Membrane Disruption

To test if membrane disruption played a role in the potentiation of antibiotics seen in Table 1, the possibility of membrane disruption caused by TXA09155 was examined. Membrane disruption was assayed with two approaches: (1) a flow cytometry-based propidium iodide (PI) assay to monitor inner membrane permeabilization, and (2) a nitrocefin (NCF) assay to monitor outer membrane permeabilization. NCF is a chromogenic cephalosporin that changes from yellow to red when the amide bond in the β-lactam ring is hydrolyzed by a β-lactamase. The rate of hydrolysis in intact cells is slow, as it is limited by the rate of diffusion of periplasmic β-lactamase across the outer membrane. However, in the presence of an agent that permeabilizes the outer membrane, the rate of hydrolysis will increase [33]. The impact of TXA09155 on NCF hydrolysis is minimal at concentrations below 25 μg/mL, suggesting that TXA09155 does not interact with the outer membrane of *P. aeruginosa* at these concentrations (Figure 2A). This result is comparable to what was previously reported for TXA01182 [29] suggesting that incorporation of the *N*-2-(4-aminomethylpyrrolidin-2-yl) methyl moiety that produced TXA09155 did not introduce membrane disruption properties to the compound. Polymyxin B was used as a positive control (Figure 2B). TXA09155 did not disrupt the bacterial inner membrane below concentrations of 50 μg/mL compared with water and polymyxin B as vehicle and positive controls, respectively (Figure 2C). In the PI assay, log-phase *P. aeruginosa* cells were mixed with various concentrations of TXA09155, followed by the addition of PI. Cells whose membranes remain intact exclude PI and remain non-fluorescent, while cells with compromised membrane integrity allow PI to enter and bind to DNA, resulting in fluorescence.

### 2.3. TXA09155 Inhibits the Efflux of Ethidium Bromide and Levofloxacin

The ability of TXA09155 to inhibit efflux in *P. aeruginosa* was studied using two quantitative assays to establish efflux inhibition as the MOA. In the first assay, the efflux of ethidium bromide (EtBr) by *P. aeruginosa* cells is studied in the presence of different concentrations of TXA09155. Wild-type *P. aeruginosa* ATCC 27853 cells were incubated with EtBr to allow for the intracellular accumulation and were treated with CCCP to inhibit active efflux. When bound to intracellular bacterial DNA, EtBr fluoresces brightly, while any unbound EtBr outside bacterial cells exhibits little or no fluorescence. Following activation by the addition of glucose, the efflux of EtBr can be followed in real time as a decrease in fluorescence based on the concentration of TXA09155. As seen in Figure 3A, the fluorescence intensity increased proportionally with the increasing concentration of TXA09155, indicating intracellular accumulation of EtBr and supporting a role in efflux inhibition by TXA09155.

The second assay quantifies the levels of levofloxacin accumulated inside *P. aeruginosa* after treatment with different concentrations of TXA09155. *P. aeruginosa* DA7232-harboring mutations in DNA gyrase (*gyrA*-T83I) and topoisomerase IV (*parC*-S80L) were used for this study since they are highly resistant to levofloxacin and thus can tolerate and accumulate high concentrations of the antimicrobial [34]. The levofloxacin MIC for this strain decreased from 256 to 1 µg/mL in the presence of 6.25 µg/mL TXA09155, making this strain ideal for the study. In the assay, *P. aeruginosa* DA7232 cells are incubated with levofloxacin and TXA09155 to allow for intracellular accumulation. After TXA09155 removal and membrane permeabilization, the levels of accumulated levofloxacin can be quantified fluorescently. Similar assays have been described elsewhere with the same purpose [35,36]. TXA09155 led to the accumulation of levofloxacin inside *P. aeruginosa* in a concentration-dependent manner (Figure 3B) in support of a role in efflux inhibition.

### 2.4. TXA09155 Does Not Affect the Proton Gradient across P. aeruginosa Inner Membrane

RND efflux pumps couple the influx of protons from the proton gradient across the inner membrane to the efflux of drugs and other molecules [37]. Therefore, it is important to verify that the MOA by which TXA09155 potentiates the antimicrobials mentioned in Table 1 does not involve disruption of the proton gradient. The effect of TXA09155 on the proton gradient of *P. aeruginosa* was assessed with a fluorescence-based 3,3′-diethyloxacarbocyanine iodide [DiOC_2_ (3)] assay that was developed to measure membrane potential changes in Gram-negative bacteria [38]. DiOC_2_ (3) is a positively charged fluorescent dye that accumulates within cells in a charge-dependent manner, shifting its fluorescence spectrum from green to red at high concentrations due to concentration-dependent dye stacking [39]. As shown in Figure 4A, treatment of *P. aeruginosa* with TXA09155 at concentrations ranging from 20 to 0.01 μM (20 μM = 7.32 µg/mL TXA09155) did not lead to membrane depolarization, ruling this out as the MOA responsible for antibiotic potentiation. Azithromycin and CCCP were used as positive and negative controls, respectively.

### 2.5. TXA09155 Does Not Deplete ATP Levels in P. aeruginosa

Another approach to assess any effect a drug may have on membrane integrity is to monitor ATP levels following drug treatment. Membrane disruption leads to membrane depolarization and impairment of the proton motive force (PMF), which is essential for a variety of critical bacterial processes, such as ATP synthesis [40]. As shown in Figure 4B, treatment of *P. aeruginosa* with TXA09155 did not result in ATP depletion. CCCP was used as positive control.

### 2.6. TXA09155 Is Active in P. aeruginosa Mutants Overexpressing MexAB-OprM and MexXY-OprM Efflux Pumps

Next, the ability of TXA09155 to inhibit *P. aeruginosa* mutants overexpressing MexAB-OprM and MexXY-OprM was evaluated. Loss of function mutations in *nalB* and *mexZ* resulting in overexpression of MexAB-OprM and MexXY-OprM, respectively, have been identified in *P. aeruginosa* clinical isolates and are associated with resistance to cephalosporins, fluoroquinolones, and aminoglycosides [41,42,43,44,45,46,47]. An EPI should reverse the antimicrobial susceptibility lost in bacteria overexpressing efflux pumps. Therefore, the EPI activity of TXA09155 was studied using antibiotics that display efflux liabilities in these two RND efflux pumps (Table 3 and Appendix A). The level of cefpirome and levofloxacin potentiation seen with TXA09155 in MexAB-OprM and MexXY-OprM overproducing strains K1455 and K2415, respectively was comparable to the effect of inactivating both pumps by deleting *oprM* (strain K3698). The level of cotrimoxazole, doxycycline, minocycline, and chloramphenicol potentiation by TXA09155 was greater than the effect seen by inactivating MexAB-OprM and MexXY-OprM (compare K1455/K3698 ratios to K1455 ± TXA09155 and K2415 ± TXA09155). In agreement with these observations, TXA09155 did not significantly potentiate cefpirome or levofloxacin in the MexAB-OprM and MexXY-OprM-deficient strain K3698, but potentiated cotrimoxazole, doxycycline, minocycline, and chloramphenicol significantly in the same strain. Doxycycline and minocycline belong to the tetracycline class of antibiotics and thus inhibit protein synthesis through reversible binding to bacterial 30S ribosomal subunits [48]. Chloramphenicol is a broad-spectrum antibiotic that reversibly binds to the bacterial 50S ribosomal subunit inhibiting protein synthesis [49]. Cotrimoxazole is the antibacterial combination product of trimethoprim and sulfamethoxazole. These components inhibit enzyme systems involved in the bacterial synthesis of tetrahydrofolic acid and block the initiation of protein synthesis among other cellular processes [50,51,52,53]. Together, these data suggest that in addition to inhibiting efflux in MexAB-OprM and MexXY-OprM overproducing strains, TXA09155 may have a secondary MOA synergistic with protein synthesis inhibitors. Finally, imipenem, which is not the substrate of RND efflux pumps in *P. aeruginosa*, was not potentiated by TXA09155 in any of the strains tested.

### 2.7. TXA09155 Potentiates Levofloxacin in P. aeruginosa Clinical Isolates from the United States and Other Countries

The ability of TXA09155 to potentiate levofloxacin in *P. aeruginosa* clinical isolates was tested in two outsourced studies. For both studies, TXA09155 was tested at a fixed concentration of 6.25 μg/mL. In the first study performed by Micromyx (Kalamazoo, MI, USA), 209 recent *P. aeruginosa* clinical isolates for which the levofloxacin MIC ranged between 0.03 to >32 μg/mL were collected from three hospitals across the US. Isolate selection was unbiased for resistance mechanisms. In this panel, TXA09155 decreased the levofloxacin MIC at which 50% of strains are inhibited (MIC_50_) from 1 to 0.12 μg/mL and decreased the MIC_90_ from 16 to 2 μg/mL (Figure 5A and Appendix A). In the second study performed by IHMA (Switzerland), 300 recent MDR *P. aeruginosa* clinical isolates for which the levofloxacin MIC ranged between 0.06 to 64 μg/mL were collected from hospitals across 63 countries. In this panel, TXA09155 decreased the levofloxacin MIC_50_ from 0.5 to 0.06 μg/mL and decreased the MIC_90_ from 32 to 8 μg/mL (Figure 5B and Appendix A). In both studies, the percentage of levofloxacin-resistant isolates dropped by 24% and 10%, respectively, highlighting the potential the TXA09155 scaffold has at reducing resistance and increasing susceptibility to antibiotics affected by efflux, such as levofloxacin.

### 2.8. TXA09155 Shows Superior Potency against Multidrug-Resistant P. aeruginosa Clinical Isolates from the CDC-FDA

The significant levofloxacin potentiation displayed in the previous *P. aeruginosa* panels containing resistant isolates was a promising sign that suggested that TXA09155 might retain its activity against MDR *P. aeruginosa*. Therefore, the EPI activity of TXA09155 was further tested against 34 MDR *P. aeruginosa* clinical isolates obtained from the CDC & FDA Antibiotic Resistance (AR) Isolate Bank [54]. The genomes of these MDR isolates have been sequenced and include mutations associated with MexAB-OprM overexpression (*nalC*-G71E and *mexR-*V126Q), mutations that lead to fluoroquinolone resistance (*gyrA*-T83I and *gyrA*-T133H), as well as other known resistant mechanisms (Appendix A) [34,55,56,57,58,59,60,61]. The levofloxacin MIC in these 34 MDR isolates ranged from 0.5 to 128 μg/mL. Again, TXA09155 was tested at a fixed concentration of 6.25 μg/mL. TXA09155 was able to decrease the levofloxacin MIC_50_ from 32 to 2 μg/mL and decrease the MIC_90_ from 64 to 8 μg/mL (Figure 5C and Appendix A). More importantly, the TXA09155/levofloxacin combination lowered the percentage of levofloxacin-resistant isolates from 85% to 35%. The activity of TXA09155 outperformed the activity of its parent compound TXA01182 (24), which lowered the percentage of levofloxacin-resistant isolates from 85% to 56% when combined with levofloxacin, again confirming that constraining the diamine side chain of TXA01182 resulted in a more active EPI. MC-04,124, a known EPI active against *P. aeruginosa* [62], failed to decrease the MIC_50_, MIC_90_, and levofloxacin-resistance percentage when tested at the same concentration as TXA09155 and TXA01182, which highlights the superior potency of the TAXIS EPI scaffolds.

### 2.9. TXA09155 Lowers the Frequency of Resistance to Levofloxacin

In addition to reducing the levels of intrinsic resistance, a potent EPI is also expected to significantly reverse acquired resistance as well as decrease the frequency at which antimicrobial resistance emerges. To test whether TXA09155 increases the selective pressure on resistant mutant emergence when combined with antimicrobials, the effect of TXA09155 on the frequency of resistance (FoR) of *P. aeruginosa* ATCC 27853 to levofloxacin was tested. TXA09155 reduced the FoR to levofloxacin more than 149-fold (Table 4). The undetectable levels of resistance seen in the TXA09155/levofloxacin combination would be of great value in a clinical setting, particularly in cystic fibrosis patients infected with *P. aeruginosa*, where patients are colonized by hypermutable strains that persist for years [63].

### 2.10. Genes Involved in Resistance to TXA09155 Alone or Combined with Levofloxacin

A study was conducted to understand how *P. aeruginosa* becomes resistant to TXA09155, alone or in combination with levofloxacin. To understand resistance to TXA09155 alone, *P. aeruginosa* ATCC 27853 was exposed to four times or one time the MIC of TXA09155 (200 or 50 μg/mL, respectively). All resistant colonies were verified by determining the MIC to TXA09155. Resistance to TXA09155 did not result in cross-resistance to other known antibiotics (Appendix A). Resistance to four times the MIC of TXA09155 alone arose at a frequency of 1.40 × 10^−6^ and resulted in single point mutations in the *phoQ* gene, the sensor kinase component of the two-component regulatory system PhoP-PhoQ (Table 5, strain EPIR1S, EPIR9S, and EPIR20L). This two-component system regulates the expression of more than 100 genes in *P. aeruginosa*, including genes that code for efflux components (*mexX*), and plays a role in the resistance to cationic antimicrobial peptides and aminoglycoside antibiotics [64,65].

Resistance to one time the MIC of TXA09155 alone occurred at a frequency of 1.9 × 10^−3^ and resulted in a single point mutation that introduced an early stop codon in the *ompH* gene (strain EPIR43). OmpH is a homolog of outer membrane chaperone Skp [66]. It has been found to be overexpressed in *P. aeruginosa* isolates resistant to ampicillin and kanamycin [67]. Deletion of *ompH* leads to hyper susceptibility to ertapenem, ceftazidime, levofloxacin, tigecycline, and cotrimoxazole [68]. In agreement with these findings, EPIR43 gained susceptibility to levofloxacin, ceftazidime, tigecycline, doxycycline, meropenem, amikacin, and azithromycin (Appendix A). The level of susceptibility gained to levofloxacin and doxycycline matches the level of potentiation observed with the lowest active concentration of TXA09155 (Table 1). In *E. coli*, more than 30 envelope proteins have been found to interact with the Skp chaperone [69,70]. *E. coli 10omph* genetically interacts with AcrD, a component of the AcrAD-TolC aminoglycoside efflux pump [71,72]. The amino acid sequence of AcrD shares 62% similarity with *P. aeruginosa* MexB. Thus, it is possible that in *P. aeruginosa*, OmpH is involved in folding an efflux pump component targeted by TXA09155. If so, loss of function of *ompH* in *P. aeruginosa* could result in misfolded efflux pump components and therefore diminished antibiotic efflux leading to hypersensitivity. Alternatively, TXA09155 could be directly inhibiting the function of OmpH, resulting in misfolded efflux pump components and antibiotic hypersusceptibility.

Resistance to the TXA09155/levofloxacin combination was studied in the *P. aeruginosa* strain DA7232, which as mentioned above, is resistant to levofloxacin due to its DNA gyrase and Topoisomerase 4 mutations [34]. Since this strain is resistant to fluoroquinolones, it was hypothesized that resistance would arise predominantly from mutations related to efflux or additional mechanisms of action in this genetic background. *P. aeruginosa* DA7232 was exposed to 3.13 μg/mL of TXA09155 (lowest concentration that potentiated levofloxacin eight-fold) and 256 μg/mL of levofloxacin (1X-MIC) to obtain resistant colonies. Resistance to the combination occurred at a frequency of 2.48 × 10^−8^ and resulted in single point mutations in the tryptophan-tRNA ligase gene, *trpS* (strain EPIR24L), suggesting a possible role in protein synthesis inhibition for TXA09155. If protein synthesis inhibition is a secondary mechanism of action of TXA09155, this would explain why it shows synergy with ribosome inhibitors doxycycline, minocycline, and chloramphenicol in efflux-deficient strain K3698 (Table 2).

### 2.11. Time-Kill Assay

In addition to assessing the potentiation activity of TXA09155 in vitro, its potentiation of a minimally bactericidal concentration of moxifloxacin (1×-MIC) was probed against *P. aeruginosa* ATCC 27853 with time-kill studies. Figure 6 shows time-kill curves with moxifloxacin alone or combined with TXA09155. By itself, TXA09155 did not affect the growth of *P. aeruginosa* ATCC 27853 at the highest concentration tested (gray curve). TXA09155 enhanced moxifloxacin’s killing kinetics in a concentration-dependent manner (green, purple and black curves). After 3 h of incubation, the combinations killed more bacteria by a magnitude of ≥3 logs compared to moxifloxacin alone (orange curve). After 6 h of incubation, the combinations increased bacterial killing by a magnitude of ≥3.9 logs compared to moxifloxacin alone. At 24 h, the combinations achieved ≥ 4 logs of killing more than moxifloxacin alone. These results suggest that the killing kinetics for the combination of TXA09155/moxifloxacin are faster than those of moxifloxacin alone. The potentiation of a minimally bactericidal concentration of moxifloxacin (1×-MIC) by TXA09155 seen in Figure 6 is consistent with similar studies performed with a TXA01182/levofloxacin combination in *P. aeruginosa* [29].

### 2.12. TXA09155 Has a Favorable Physiochemical and ADME Profile

Along with its microbiological evaluation, TXA09155 was evaluated for its physiochemical and in vitro ADME properties. In general, TXA09155 follows Lipinski’s rule-of-five, a rule describing molecular properties believed to be important for a drug’s pharmacokinetics in the human body, which is highly soluble (>155 μM at pH 7.4). Table 6 summarizes the in vitro chemical absorption, distribution, metabolism, excretion, and toxicity (ADMET) profile of TXA09155. Overall, the metabolic stability of TXA09155 is good across two species (human and rat). It did not inhibit cytochrome P450 isoforms having IC_50_ values greater than 100 μM for CYP1A2, CYP2C19, CYP2C9, and 67.2 and 28.5 μM for CYP2D6 and CYP3A4, respectively. The observed cytotoxicity IC_50_ values of TXA09155 in CellTiterGlo^TM^ assay (293T, A549) were all greater than 40 μM. Additionally, TXA09155 did not display hemolytic activity in human RBCs in the 12.5 to 100 µM concentrations range. The only safety concern for TXA09155 was its hERG liability, which has an IC_50_ of 16 μM in the patch clamp assay.

## 3. Materials and methods

### 3.1. Synthesis Reagents, Analytical Instruments and Reaction Conditions

All reactions, unless otherwise stated, were conducted under nitrogen atmosphere. Reaction monitoring and follow-up were performed using aluminum backed Silica G TLC plates (UV254 e Sorbent Technologies, Norcross, Georgia), visualized with ultraviolet light. Flash column chromatography was performed on a Combi Flash Rf Teledyne ISCO using hexane, ethyl acetate, dichloromethane, and methanol. The ^1^H (300 MHz) spectra were performed in CDCl_3_, Methanol-d_4_, and DMSO-d_6_ and recorded on a Varian Oxford 300 MHz Multinuclear NMR Spectrometer. Data are expressed in parts per million relative to the residual nondeuterated solvent signals; spin multiplicities are given as s (singlet), d (doublet), dd (doublet of doublets), t (triplet), dt (doublet of triplets), q (quartet), m (multiplet), bs (broad singlet), and bt (broad triplet); and coupling constants (J) are reported in Hertz. The final compound tested for biological activity was analyzed for purity using Shimadzu LCMS-2020 system monitoring absorbances at both 254 and 280 nm. All chemical reagents used in this manuscript were purchased either from Sigma-Aldrich (St. Louis, MS, USA) or from Combi-Blocks Inc. (San Diego, CA, USA).

The synthetic scheme for the preparation of TXA09155 is outlined in Figure 7. In order to maintain the same C2-amine stereochemistry as in TXA01182 (*S*-stereochemistry), the synthesis started from commercially available (2*S*,4*R*)-4-hydroxypyrrolidine-2-carboxylic acid **1**. For initial proof of concept regarding gain of activity with a conformationally constrained scaffold, the stereochemistry of the 4-hydroxy group was not taken into consideration at this stage. Esterification of **1** followed by benzylation afforded intermediate compound **3**. Tosylation of the hydroxy group and then replacing it with the CN-group changed the stereochemistry at the C4. Reduction of the methyl ester of **5** with LiAlH_4_ afforded the hydroxyl methyl derivative **6**. Two-step Mitsunobu protocol was then followed to convert the hydroxyl group of **6** to the amine group as shown in compound **8** in 87% yield over two steps. Coupling of **8** with known indole acid **9** afforded the amide compound **10** in 60% yield. Nitrile reduction and debenzylation were achieved by two-step hydrogenation procedure using Raney-Ni and Pd/C as catalysts. Finally, TXA09155 was synthesized as a HCl salt in 64% yield over 3 steps.

Step (a): Synthesis of methyl (*2S*,*4R*)-4-hydroxypyrrolidine-2-carboxylate (**2**): Commercially available (*2S*,*4R*)-4-hydroxypyrrolidine-2-carboxylic acid (30 g, 0.23 mol) in MeOH (300 mL) was cooled to 0 °C, SOCl_2_ (33 mL) was added portion wise over 10 min. The resulting mixture was stirred at room temperature overnight. The methanol was removed, and the residue was triturated with CH_2_Cl_2_ (200 mL) to give the desired product as a white solid, which was used in the next step without further purification.

Step (b): Synthesis of methyl (*2S*,*4R*)-1-benzyl-4-hydroxypyrrolidine-2-carboxylate (**3**): To the crude methyl (*2S*,*4R*)-4-hydroxypyrrolidine-2-carboxylate (~0.23 mol) in CH_2_Cl_2_ (300 mL) was added TEA (96 mL, 0.69 mol) and then BnBr (32.6 mL, 0.27 mol) portion wise. After stirring at room temperature overnight, the reaction mixture was washed with water, 1N NaOH solution, brine, dried over Na_2_SO_4_, and concentrated to give a crude product, which was purified by silica gel plug. Elution with 50% EtOAc/hexanes afforded the desired product (50 g, 86% yield) as a light-yellow liquid. ^1^H NMR (300 MHz, CDCl_3_) 7.32–7.30 (m, 5H), 4.48–4.41 (m, 1H), 3.89 (d, *J* = 13 Hz, 1H), 3.72–3.60 (m, 5H), 3.32 (dd, *J* = 10, 5 Hz, 1H), 2.46 (dd, *J* = 10, 4 Hz, 1H), 2.29–2.20 (m, 1H), 2.11–2.04 (m, 1H), 1.91 (br. s, 1H).

Step (c): Synthesis of methyl (*2S*,*4R*)-1-benzyl-4-(tosyloxy)pyrrolidine-2-carboxylate (**4**): Methyl *(2S*,*4R*)-1-benzyl-4-hydroxypyrrolidine-2-carboxylate (3) (3.25 g, 13.8 mmol) in dry pyridine (7.0 mL) was cooled to 0 °C, and TsCl (2.75 g, 14.5 mmol) was added. The resulting mixture was stirred at room temperature overnight. The reaction mixture was diluted with CH_2_Cl_2_ and washed with 10% citric acid (50 mL × 2), brine, and dried over anhydrous Na_2_SO_4_. The organic solution was concentrated to give the crude product (2.0 g, 70% yield) as a light brown oil, which was used in next step without further purification. ^1^H NMR (300 MHz, CDCl_3_) 7.77–7.73 (m, 2H), 7.33–7.22 (m, 7H), 4.98–4.95 (m, 1H), 3.84 (d, *J* = 13 Hz, 1H), 3.64 (s, 3H), 3.61–3.53 (m, 2H), 3.27–3.24 (m, 1H), 2.63 (dd, *J* = 11, 4 Hz, 1H), 2.44 (s, 3H), 2.77–2.25 (m, 2H).

Step (d): Synthesis of methyl (*2S*,*4S*)-1-benzyl-4-cyanopyrrolidine-2-carboxylate (**5**): NaCN (0.96 g, 19.5 mmol) was added to methyl (*2S*,*4R*)-1-benzyl-4-(tosyloxy)pyrrolidine-2-carboxylate (3.79 g, 9.7 mmol) in dry DMSO (10 mL). The reaction mixture was stirred at 60 °C overnight. The reaction mixture was diluted with CH_2_Cl_2_, washed with water, brine, and dried over anhydrous Na_2_SO_4_. The organic solution was concentrated to give the crude product (3.79 g, 84% yield) as a light brown oil, which was used in next step without further purification. ^1^H NMR (300 MHz, CDCl_3_) 7.31 (m, 5H), 3.95 (d, *J* = 14 Hz, 1H), 3.74 (s, 3H), 3.63–3.61 (m, 1H), 3.48 (dd, *J* = 9, 6 Hz, 1H), 3.24 (dd, *J* = 9, 5 Hz, 1H), 3.08–3.05 (m, 1H), 2.85 (dd, *J* = 9, 8 Hz, 1H), 2.56–2.31 (m, 2H).

Step (e): Synthesis of (*3S*,*5S*)-1-benzyl-5-(hydroxymethyl)pyrrolidine-3-carbonitrile (**6**): LiBH_4_ (0.36 g, 16.4 mmol) was added to methyl (*2S*,*4S*)-1-benzyl-4-cyanopyrrolidine-2-carboxylate (2.0 g, 8.2 mmol) in dry THF (30 mL). The reaction mixture was stirred at 80 °C for 1 h. The reaction mixture was cooled to room temperature, and acetone (1 mL) was added to quench the excess LiBH4. After stirring for 30 min, the solvent was removed, and the residue was partitioned between EtOAc and saturated NaHCO_3_ solution. The organic layer was washed with brine, dried over Na_2_SO_4_, concentrated, and purified on a silica gel column. Elution with 50% EtOAc/hexanes afforded the desired product (1.1 g, 62% yield) as a light-yellow oil. ^1^H NMR (300 MHz, CDCl_3_) 7.33–7.26 (m, 5H), 4.03 (d, *J* = 14 Hz, 1H), 3.77 (dd, *J* = 11, 3 Hz, 1H), 3.53–3.47 (m, 1H), 3.39 (d, *J* = 14 Hz, 1H), 3.22 (d, *J* = 10, 4 Hz, 1H), 2.99–2.94 (m, 1H), 2.85–2.79 (m, 1H), 2.61–2.56 (m, 1H), 2.40–2.23 (m, 2H).

Step (f): Synthesis of (*3S*,*5S*)-1-benzyl-5-((1,3-dioxoisoindolin-2-yl)methyl)pyrrolidine-3-carbonitrile (**7**): Phthalimide (823 mg, 5.6 mmol), Ph_3_P (1.46 g, 5.6 mmol) were added to a solution of (*3S*,*5S*)-1-benzyl-5-(hydroxymethyl)pyrrolidine-3-carbonitrile (1.1 g, 5.1 mmol) in THF (15 mL). The solution was cooled to 0 °C and DIAD (1.13 mL, 5.6 mmol) was added portion wise over 5 min. The resulting mixture was stirred at room temperature overnight. The solvent was removed, and the residue was purified on silica gel column. Elution with 10–30% EtOAc/hexanes afforded the crude product (2 g, ~90% pure), which was directly used in next step. ^1^H NMR (300 MHz, CDCl_3_) 7.87–7.84 (m, 2H), 7.76–7.71 (m, 2H), 7.32–7.19 (m, 5H), 4.23 (d, *J* = 13 Hz, 1H), 3.87 (d, *J* = 5 Hz, 2H), 3.44 (d, *J* = 14 Hz, 1H), 3.18 (dd, *J* = 10, 4 Hz, 1H), 3.03–2.96 (m, 1H), 2.94–2.85 (m, 1H), 2.56 (dd, *J* = 10, 7 Hz, 1H), 2.38–2.16 (m, 2H).

Step (g): Synthesis of (*3S*,*5S*)-5-(aminomethyl)-1-benzylpyrrolidine-3-carbonitrile (**8**): NH_2_NH_2_.H_2_O (1.5 mL) was added to the crude (*3S*,*5S*)-1-benzyl-5-((1,3-dioxoisoindolin-2-yl)methyl)pyrrolidine-3-carbonitrile from the above step (2.0 g, ~90% pure) in MeOH (30 mL). The reaction mixture as stirred at 50 °C for 2 h. The solvent was removed, and the residue was triturated with CH_2_Cl_2_. The white solid was removed by filtration, and the filtrate was washed with brine, dried over Na_2_SO_4_, concentrated and purified on silica gel column. Elution with 1% ammonia in 10% MeOH/ CH_2_Cl_2_ afforded the desired product (0.96 g, 87% yield in two steps) as a light-yellow liquid. ^1^H NMR (300 MHz, CDCl_3_) 7.33–7.26 (m, 5H), 4.02 (d, *J* = 14 Hz, 1H), 3.32 (d, *J* = 14 Hz, 1H), 3.18 (dd, *J* = 10, 2 Hz, 1H), 2.99–2.91 (m, 2H), 2.77–2.64 (m, 2H), 2.49 (dd, *J* = 9, 8 Hz, 1H), 2.34–2.22 (m, 1H), 2.13–2.05 (m, 1H).

Step (h): Synthesis of *N*-(((*2S*,*4S*)-1-benzyl-4-cyanopyrrolidin-2-yl)methyl)-6-(4-fluorophenyl)-1*H*-indole-2-carboxamide (**10**): DIPEA (0.35 mL, 1.97 mol) was added to the mixture of 6-(4-fluorophenyl)-1*H*-indole-2-carboxylic acid **9** (276 mg, 1.08 mmol), (*3S*,*5S*)-5-(aminomethyl)-1-benzylpyrrolidine-3-carbonitrile (212 mg, 0.99 mmol), EDC (227 mg, 1.18 mmol), HOBt (80 mg, 0.59 mmol) in DMF (5 mL). The reaction mixture was stirred at room temperature overnight. The reaction mixture was added to water dropwise with stirring, and the solid formed was collected by filtration. Air drying then silica gel column purification afforded the product (270 mg, 60% yield) as a pale-yellow solid. ^1^H NMR (300 MHz, CDCl_3_) δ 9.19 (bs, 1H), 7.73 (d, *J* = 9.0 Hz, 1H), 7.62–7.55 (m, 3H), 7.39–7.26 (m, 6H), 7.18–7.07 (m, 3H), 6.77 (m, 1H), 4.07 (d, 1H), 3.83 (m, 1H), 3.50–3.46 (m, 2H), 3.31 (d, *J* = 9.9 Hz, 1H), 3.01 (m, 2H), 2.57 (m, 1H), 2.41 (m, 1H), 2.14 (m, 1H).

Steps (i, j, k): Synthesis of *N*-(((*2S*,*4R*)-4-(aminomethyl)pyrrolidine-2-yl)methyl)-6-(4-fluorophenyl)-1*H*-indole-2-carboxamide hydrochloride salt (TXA09155): To a solution of *N*-(((*2S*,*4S*)-1-benzyl-4-cyanopyrrolidin-2-yl)methyl)-6-(4-fluorophenyl)-1H-indole-2-carboxamide (200 mg, 0.44 mmol) in THF (30 mL) was added Raney-Ni (200 mg, 50% in water) under H_2_ (55 psi) and stirred overnight. The reaction progress was monitored by LC-MS. After the reaction was completed, the catalyst was removed by passing through a Celite plug and washed with MeOH. The filtrate was concentrated to give the amine intermediate. This intermediate was dissolved in MeOH (20 mL). Pd/C (30 mg, 10% on carbon) was added then stirred under H_2_ (55 psi) overnight. After the reaction was completed, monitoring by LC-MS, the catalyst was removed by filtration, the filtrate was concentrated to give the crude product, which was purified on a C18 column. Elution with water/MeOH afforded the desired product as the free base form. The free base product was dissolved in MeOH (10 mL) and the added 4 N HCl in dioxane (0.5 mL). After stirring at room temperature for 1 h, the solvent was removed, and the residue was triturated with EtOAc to afford the desired product (124 mg, 64% yield) as an off-white solid. ^1^H NMR (300 MHz, D_2_O) δ 7.59 (d, *J* = 8.4 Hz, 1H), 7.49 (m, 3H), 7.24 (d, *J* = 8.1 Hz, 1H), 7.03 (t, *J* = 8.7 Hz, 2H), 6.95 (s, 1H), 3.71 (m, 1H), 3.51 (m, 2H), 3.39 (m, 1H), 3.01–2.87 (m, 3H), 2.57 (m, 1H), 2.31 (m, 1H), 1.40 (m, 1H). LC-MS found 367.20 [M + H^+^], calculated for [C21H24FN4O^+^] 367.19, Purity: ~98%.

### 3.2. Bacterial Strains, Media, and Reagents

*P. aeruginosa* ATCC 27853 was obtained from the American Type Culture Collection (ATCC; Manassas, VA, USA). *P. aeruginosa* multidrug-resistant isolates were obtained from the CDC and FDA Antibiotic Resistance Isolate Bank (Atlanta, GA, USA) [54]. *P. aeruginosa* DA7232 (*gyrA*-T83I, *parC*-S80L) was a kind gift from Prof. Dan I. Andersson, Uppsala University, Uppsala, Sweden, and has been characterized elsewhere [34]. *P. aeruginosa* strains K767 (WT), K1455 *(mexAB-oprM* overexpressed), K2415 (*mexXY-oprM* overexpressed), and K3698 (*oprM*∆) were obtained from Prof. Keith Poole, Queen’s University, Kingston, Ontario, Canada and have been characterized elsewhere [73,74]. Bacterial cells were grown in cation-adjusted Mueller Hinton (CAMH) media, brain heart infusion broth (BHI), or tryptic soy agar (TSA) plates were all obtained from Becton, Dickinson, and Company (BD; Franklin Lakes, NJ, USA). Aztreonam, ceftazidime, moxifloxacin, levofloxacin, minocycline, tigecycline, chloramphenicol, nitrocefin, and imipenem were purchased from TOKU-E (Bellingham, WA, USA). Azithromycin was purchased from Tokyo Chemical Industry (Portland, OR, USA). Cotrimoxazole was purchased from Toronto Research Chemicals (Toronto, ON, Canada). Doxycycline and polymyxin B were purchased from Sigma-Aldrich (St. Louis, MO, USA). Ethidium bromide (EtBr) and glucose were purchased from VWR (Radnor, PA, USA). MC-04,124 and TXA01182 were synthesized at TAXIS Pharmaceuticals. Carbonyl cyanide 3-chlorophenylhydrazone (CCCP) was purchased from Enzo Life Sciences (Farmingdale, NY, USA). DiOC_2_ (3) and propidium iodide were obtained from Thermo Fisher Scientific (Waltham, MA, USA).

### 3.3. Micromyx and IHMA Clinical Isolates for Susceptibility Studies

First, 209 clinical isolates of *P. aeruginosa* were collected, unbiased of susceptibility, from three US hospitals by Micromyx (now part of Microbiologics; Kalamazoo, MI, USA). IHMA (Monthey, Switzerland) collected 300 *P. aeruginosa* clinical isolates, also unbiased of susceptibility, from the following 63 countries: Argentina, Australia, Belgium, Brazil, Canada, Chile, Colombia, Costa Rica, Croatia, Czech Republic, Denmark, Dominican Republic, Ecuador, France, Georgia, Germany, Greece, Guatemala, Hong Kong, Hungary, India, Ireland, Israel, Italy, Japan, Jordan, Kazakhstan, Kenya, South Korea, Kuwait, Latvia, Lebanon, Lithuania, Malaysia, Mexico, Morocco, Netherlands, New Zealand, Nigeria, Panama, Philippines, Poland, Portugal, Puerto Rico, Qatar, Romania, Russia, Saudi Arabia, Serbia, Slovenia, South Africa, Spain, Sweden, Switzerland, Taiwan, Thailand, Tunisia, Turkey, Ukraine, United Kingdom, United States, Venezuela, and Vietnam. Isolates came from the following types of infection: genitourinary (urine), gastrointestinal (stomach), gastrointestinal (pancreas), intermittent needle therapy (wound), respiratory (sputum), chorionic villus sampling (blood), respiratory (bronchoalveolar lavage), intermittent needle therapy (burn), respiratory (endotracheal aspirate), bodily fluids (abscess/pus), intermittent needle therapy (impetiginous lesions), gastrointestinal (abscess) and respiratory (bronchial brushing). CF patients were not excluded.

### 3.4. Minimum Inhibitory Concentration (MIC) Assay for Potentiation of Antimicrobial Activity against P. aeruginosa

MIC assays were performed as described previously [29].

### 3.5. Flow Cytometry Assay for Permeabilization of Inner Cell Membranes to Propidium Iodide (PI) in P. aeruginosa

A flow cytometry assay used for assessing potential inner membranes permeabilization of *P. aeruginosa* bacterial cells to PI was conducted using the LIVE/DEAD BacLight Kit from Invitrogen (Waltham, MA, USA). Briefly, log-phase *P. aeruginosa* ATCC 27853 bacterial cells grown in BHI broth were diluted 5-fold in PBSM (1X PBS, 1MgCl_2_) to an approximate concentration of 6.5 × 10^7^ CFU/mL. The bacteria were aliquoted into tubes and mixed with TXA09155 at concentrations ranging from 1 to 1/8th times the MIC (50 to 6.25 μg/mL). Water alone was used as solvent control. Polymyxin B was used as a positive control. Intracellular PI fluorescence was detected by flow cytometry using a CytoFlex (Beckman Coulter Inc., Brea, CA, USA). The 488 nm laser was used for excitation, with the PC5.5 and FITC channels being used for emission. For each sample, the fluorescence of 10,000 individual bacterial cells was measured, and the percent of cells that stained positive for PI fluorescence was calculated.

### 3.6. Nitrocefin Cellular Assay for Outer Cell Membrane Permeabilization Assessment in P. aeruginosa

Assessment of outer membrane permeabilization was carried out as described previously [29].

### 3.7. Fluorescence-Based Cellular Assay for Inhibition of Pump-Mediated Efflux of Ethidium Bromide (EtBr)

Efflux of EtBr from *P. aeruginosa* in the presence of TXA09155 was carried out as described previously [29].

### 3.8. Levofloxacin Accumulation Assay

*P. aeruginosa* DA7232 was grown to an OD_600_ of 0.6 and treated with levofloxacin (64 μg/mL) alone or in combination with compounds at sub-inhibitory concentrations (1/2, 1/4th and 1/8th MIC). The treated bacterial culture samples were incubated on ice for 15 min, centrifuged, washed with 1X PBS (pH 7.4), and then resuspended in 1 mL of glycine-HCl buffer (pH 3.0) overnight, followed by centrifugation. The fluorescence of 100 μL of supernatant was read at 490 nm following a 355 nm excitation in a SpectraMax iD5 spectrophotometer (Molecular Devices, San Jose, CA, USA).

### 3.9. Membrane Polarization Assay

*P. aeruginosa* membrane polarization assays were conducted as described previously [38] using strain ATCC 27853. Briefly, exponentially grown *P. aeruginosa* culture (OD_600_ of 0.7–0.8) was harvested by centrifugation, and the pellet was resuspended to an OD_600_ of 1.0 in sodium phosphate buffer (0.13 M NaCl, 7 mM Na_2_HPO_4_, 3 mM NaH_2_PO_4_; pH 7.0). To minimize the exclusion of DiOC_2_ (3) dye, bacteria were treated with 10 mM EDTA for 5 min, centrifuged and resuspended to a final OD_600_ of 1.0 in the resuspension buffer (130 mM NaCl, 60 mM Na_2_HPO_4_, 60 mM NaH_2_PO_4_, 10 mM glucose, 5 mM KCl, 0.5 mM MgCl_2_). Cells were loaded with 30 μM DiOC_2_ (3) and added to a 96-well opaque microplate for a final volume of 200 μL in each well. Prior to the addition of cells, an indicated compound and the controls were added to the wells. Compounds were incubated with DiOC_2_ (3) loaded cells for 30 min at 37 °C. DiOC_2_ (3) fluorescence was recorded in a microplate reader using 450 nm excitation and 670 nm emission (for red fluorescence). Background correction for each concentration of each compound was obtained from reading the fluorescence of compounds in the resuspension buffer without cells. After background correction, fluorescence was normalized to that of the vehicle (DMSO) only. After a 30 min incubation at 37 °C, DiOC_2_ (3) fluorescence was recorded using a SpectraMax iD5 spectrophotometer (Molecular Devices, San Jose, CA, USA) following excitation at 450 nm. Red fluorescence intensity was recorded at 670 nm emission.

### 3.10. Determination of Intracellular ATP Levels

*P. aeruginosa* intracellular ATP levels were determined as described previously [36] using an ATP determination kit (catalog number A22066; Thermo Fisher Scientific, Waltham, MA, USA). A bacterial culture of strain ATCC 27853 was grown to the mid-log phase (OD_600_ = 0.7), washed, and resuspended in the same volume of PBSM (1X PBS, 1mM MgCl_2_). The bacterial culture was treated with sub-inhibitory concentrations of compounds (1/4th and 1/8th MIC) for 3 h at 37 °C. After treatment, the cells were lysed in chloroform, which was subsequently removed by boiling at 80 °C. The persistent ATP from cell lysate was measured in a 96-well black flat-bottom plate and plotted as relative luminescence units. A culture treated with CCCP (12.5 and 6.25 μg/mL) was included as positive controls. Error bars represent the standard deviation of triplicates.

### 3.11. Frequency of Resistance (FoR) Study

The frequency of resistance studies was carried out as described previously [29].

### 3.12. Whole-Genome Sequencing

DNA extraction, library preparation, and whole-genome sequencing of *P. aeruginosa* parent strains ATCC 27853 and DA7232, and single isolates EPIR1S, EPIR9S, EPIR20L, EPIR43, and EPIR24L were performed by CD Genomics (New York, NY, USA) using Illumina. Sequencing reads were mapped using the published genome of *P. aeruginosa* ATCC 27853 (GenBank accession number CP011857) or the published genome of *P. aeruginosa* PAO1 (parent strain of DA7232) as reference genomes with the BWA-MEM tool from the Galaxy web platform (https://usegalaxy.org/ last accessed on 1 August 2021.) [75]. Variations in the genomes between resistant strains and parent strains were identified using the LoFreq tool from the same platform.

### 3.13. Time-Kill Studies

Time-kill studies were carried out as described previously [29] with the following changes. When indicated, moxifloxacin was added to the prepared bacterial suspensions at one time the MIC (2 μg/mL). TXA09155 was added to bacterial suspensions at 1/8th, 1/12.5th, and 1/16th times the MIC (6.25, 4, and 3.125 μg/mL, respectively).

### 3.14. Metabolic Stability Studies and hERG Assay

The metabolic stability of TXA09155 was determined in human, mouse, and rat liver microsomes by Eurofins (St. Charles, MO, USA) using the substrate depletion method (in vitro half-life (t_1/2_) method). Quantification was performed by LC/MS/MS. In this assay, depletion of the parent compound is measured over time to determine the compound’s in vitro *t*_1/2_ and hepatic extraction ratios (ER). This is used to predict metabolic clearance. Inhibition of the hERG channel by TXA09155 was determined by Charles River Laboratories (Cleveland, OH, USA). This assay utilizes electrophysiology measurements on the electric current passing through the hERG channel heterologously expressed in a stable CHO cell line. Channels are open by a hERG-specific voltage protocol, and the compound effects are directly characterized by their block of the hERG current. The assay is performed on an automated platform, Q-Patch.

## 4. Conclusions

This work was designed to improve the activities of previously reported EPI TXA01182 by adopting the principle of constraining molecular flexibility. A common practice during the medicinal chemistry optimization of hit/lead compounds involves the application of conformation constraints since this can have a strong effect on the binding affinity of flexible molecules. Conformation constraint can be achieved by cyclization, enantiomeric substitutions, or by the introduction of stereochemically constraining chemical groups. With this in mind, the medicinal chemistry team at TAXIS Pharmaceuticals sought to constrain the diamine sidechain moiety of previously reported EPI TXA01182 by cyclization of the C2-amine and C4 positions. This resulted in TXA09155, an EPI twice as active in regard to potentiating the MIC of antibiotics from multiple classes with known efflux liabilities in wild-type *P. aeruginosa*. This gain in activity appears to be of clinical relevance since the activity TXA09155 displayed in wild-type *P. aeruginosa* translated across hundreds of clinical isolates with 334 of them being MDR isolates. The gain of activity in TXA09155 could be explained by the conformation constraint of its diamine sidechain or by a possible secondary mechanism of action. TXA09155 was able to significantly potentiate ribosome inhibitors doxycycline, minocycline, and chloramphenicol in strains deficient in MexAB-OprM and MexXY-OprM efflux pumps while not potentiating fluoroquinolones or cephalosporins in the same strain. In addition, resistance to the TXA09155/levofloxacin combination was achieved with a point mutation in the tryptophan-tRNA ligase *trpS*, suggesting a possible role in protein synthesis inhibition by TXA09155. Resistance to TXA09155 alone did not result in cross resistance to antipseudomonal antibiotics, which would be an advantage at the clinical setting where antibiotic treatment can result in multidrug resistance associated with mutations in multidrug efflux pumps [76,77]. Finally, the ADMET profile of a drug scaffold plays a key role in drug discovery and development. A high-quality drug candidate should show appropriate ADMET properties. Not surprisingly, many drug candidates fail to become drugs due to suboptimal ADMET properties. TXA09155 showed an overall good ADMET profile, suggesting that this EPI class has the potential to become a drug. A detailed SAR study and optimization effort resulting in further optimized candidates from this class will be reported next. This report will address acute toxicity, cytotoxicity, nephrotoxicity, hERG binding, PK parameters and lastly in vivo efficacy.

Although a portion of this work concentrates on TXA09155/levofloxacin activity against MDR clinical isolates, an MDR phenotype is only one of the multiple factors that affects the outcome of *P. aeruginosa* treatment at the clinic. *P. aeruginosa* possesses an overabundance of virulence determinants, including the production of biofilm, pigments, exotoxins, proteases, flagella, and secretion systems, the majority of which show no relevant correlation with an isolate’s MDR phenotype [78]. Additionally, about half of *P. aeruginosa* isolates from burn victims are carbapenem-resistant due to various mechanism [79], highlighting the potential beneficial outcome that TXA09155/levofloxacin treatment would have on these cases. Finally, time-consuming practices of pathogen identification and antibiotic susceptibility testing delay diagnosis and force clinicians to employ empirical antibiotic treatment, risking the development of antibiotic resistance [80].

## 5. Patents

Some works of this study are published in the following patent applications: LaVoie, E.; Parhi, A.; Yuan, Y.; Zhang, Y.; Sun, Y. Indole Derivatives as Efflux Pump Inhibitors. WO2018,165,611. LaVoie, E.; Parhi, A.; Zhang, Y.; Yuan, Y.; Sun, Y. Bacterial Efflux Pump Inhibitors. WO2018165612.

## Figures and Tables

**Figure 1 antibiotics-11-00716-f001:**
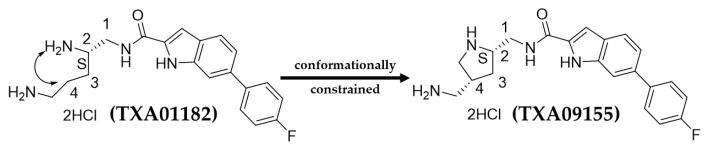
Structures of TXA01182 and TXA09155.

**Figure 2 antibiotics-11-00716-f002:**
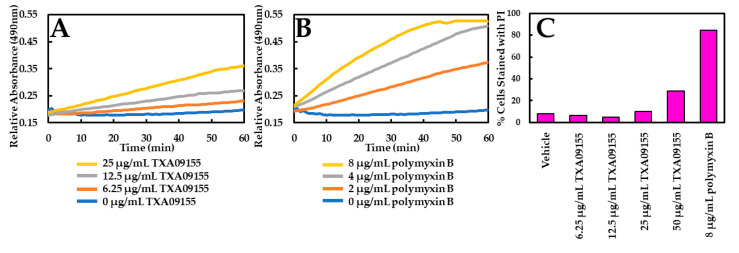
Outer- and inner-membrane permeabilization studies with TXA09155. Basal levels of NCF hydrolysis (**A**,**B**) or PI fluorescence (**C**) are observed upon addition of TXA09155 at concentrations below 12.5 and 50 μg/mL, respectively, indicating intact outer and inner membranes. Experiments were repeated at least three times, and a representative experiment is shown.

**Figure 3 antibiotics-11-00716-f003:**
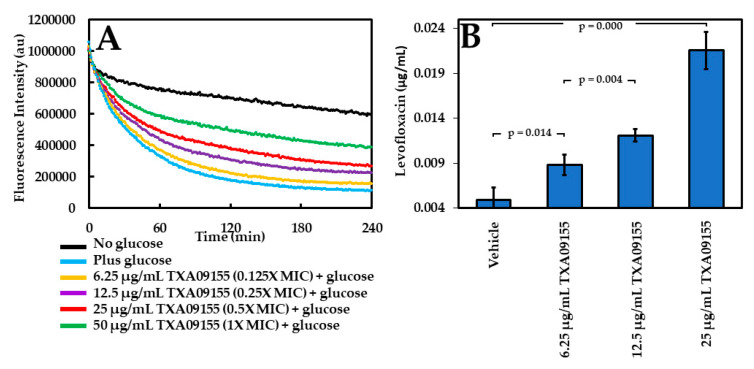
(**A**) TXA09155 concentration-dependent inhibition of EtBr efflux. (**B**) Levofloxacin accumulation in the presence of TXA09155. Experiments were repeated at least three times; a representative experiment is shown in (**A**), and values in (**B**) represent mean ± S.D.

**Figure 4 antibiotics-11-00716-f004:**
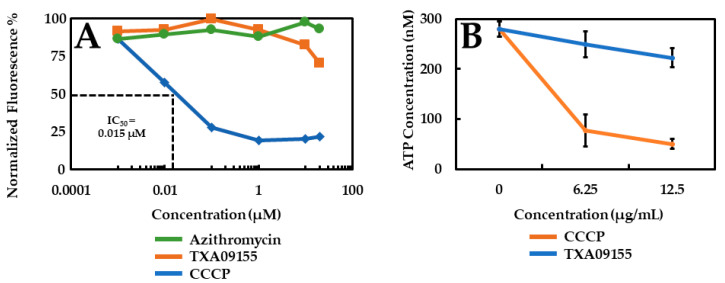
(**A**) TXA09155 did not depolarize the inner membrane of *P. aeruginosa*, which was determined by the DiOC_2_ (3) membrane polarization assay. (**B**) TXA09155 treatment of *P. aeruginosa* did not result in ATP depletion. Experiments were repeated at least three times; a representative experiment is shown in (**A**), and values represent mean ± S.D. in (**B**).

**Figure 5 antibiotics-11-00716-f005:**
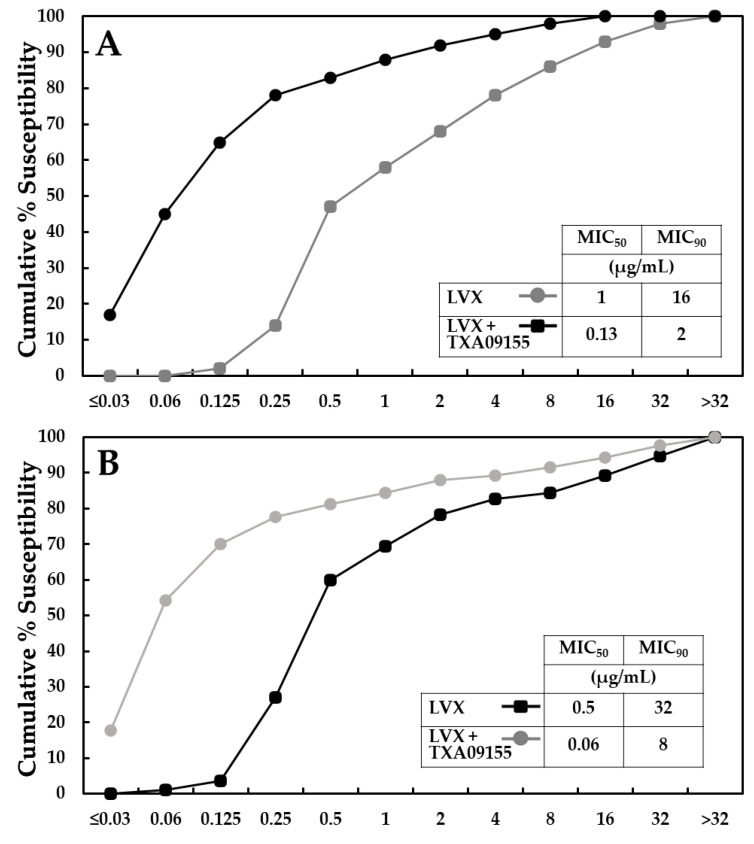
The activity of the TXA09155/levofloxacin combination against 209 clinical isolates from across the US (**A**); 300 clinical isolates from 66 countries (**B**); and 44 CDC/FDA MDR clinical isolates (**C**) of *P. aeruginosa* were determined with a fixed concentration of TXA09155 (6.25 μg/mL). MC-04,124 and TXA01182 were used as comparator EPIs at a fixed concentration of 6.25 μg/mL. Datapoints represent the percentage of strains in each panel susceptible to the indicated levofloxacin concentration.

**Figure 6 antibiotics-11-00716-f006:**
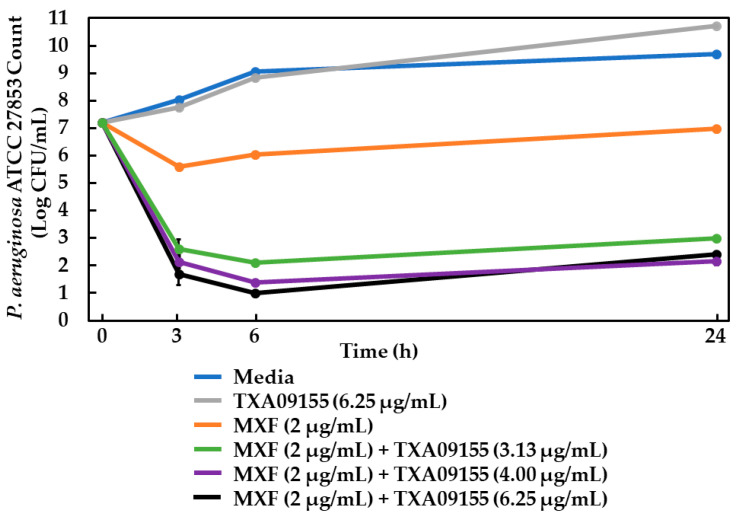
Time-kill kinetics of moxifloxacin (MXF) alone and in combination with different concentrations of TXA09155 on *P. aeruginosa*. Values expressed as mean log_10_ of CFU/mL. Error bars represent standard deviation.

**Figure 7 antibiotics-11-00716-f007:**
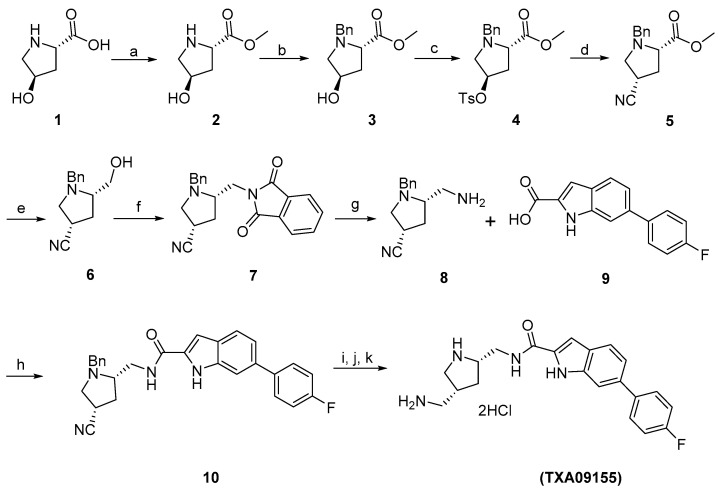
Synthetic Scheme for the preparation of TXA09155. (**a**) SOCl_2_, MeOH, 0 °C-room temp., 12 h, quantitative; (**b**) BnBr, TEA, DCM, 12 h, 86%; (**c**) TsCl, Pyr, room temp., 12 h, 70%; (**d**) NaCN, DMSO, 60 °C, 12 h, 84%: (**e**) LiBH_4_, THF, 80 °C, 1 h, 62%: (**f**) Phthalimide, PPh_3_, DIPEA, THF, room temp., 12 h, quantitative; (**g**) NH_2_NH_2._H_2_O, MeOH, 50 °C, 2 h, 87%; (**h**) EDC, HOBt, DIPEA, DMF, room temp., 12 h, 60%; (**i**) H_2_, Raney-Ni, THF; (**j**) H_2_, Pd/C, MeOH; (**k**) HCl, MeOH, 64% (over 3 steps).

**Table 1 antibiotics-11-00716-t001:** Comparison of potentiation abilities between TXA01182 and TXA09155 in wild-type *P. aeruginosa* ATCC 27853.

Antimicrobial	MICs (µg/mL)		
	Lack of EPI	+6.25 µg/mL of TXA01182 (Fold Change)	+6.25 µg/mL of TXA09155 (Fold Change)
Moxifloxacin	2	0.063 (32)	0.031 (16)
Levofloxacin	1	0.125 (8)	0.063 (16)
Doxycycline	32	2 (16)	1 (32)
Minocycline	32	1 (32)	0.5 (64)
Chloramphenicol	>256	32 (>8)	16 (>16)

**Table 2 antibiotics-11-00716-t002:** Synergistic in vitro activity of TXA09155 combined with various antibiotics against wild-type *P. aeruginosa* ATCC 27853.

Antibiotics	Antibiotic MIC (μg/mL) in Presence of TXA09155 ^‡^ at a Concentration (μg/mL) of:					E_max_ *	EC_50_ **	FIC_index_
	0	3.13	6.25	12.5	25			
Levofloxacin	1	1	0.063	0.063	0.063	16	6.25	0.188
Moxifloxacin	2	2	0.031	0.031	0.016	125	6.25	0.14
Doxycycline	32	16	1	1	1	32	6.25	0.156
Minocycline	32	16	0.5	0.25	0.125	256	6.25	0.14
Cefpirome	8	4	1	0.5	0.5	16	6.25	0.25
Aztreonam	8	4	2	0.25	0.25	32	6.25	0.125
Chloramphenicol	>256	>256	16	4	2	>128	6.25	0.187
Cotrimoxazole	256	256	8	4	2	128	6.25	0.156
Imipenem ^#^	2	2	2	2	2	1	ND	ND

^#^ Not the substrate for RND efflux pumps in *P. aeruginosa;* ^‡^ TXA09155 MIC in ATCC 27853 is 50 μg/mL; * E_max_, ratio between MIC without EPI and MIC in the presence of a maximal potentiating concentration of EPI; ** EC_50_, concentration of EPI at which half potentiating effect is achieved. FIC_index_ is the sum of the FIC_index_ of antibiotics and TXA09155. MIC 256 was used to calculate the FIC_index_ of chloramphenicol. FIC_index_ < 0.5 was considered as a drug synergistic interaction. ND, not determined.

**Table 3 antibiotics-11-00716-t003:** Comparison of antibiotic potentiation by TXA09155 with the effects that pump deletion has on MICs.

Antibiotics	MIC Ratios			
	K1455 (↑*mexAB-oprM*)/K3698 (∆*oprM*)	K1455 (↑*mexAB-oprM*) ± TXA09155 ^#^	K2415 (↑*mexXY-oprM*) ± TXA09155 ^#^	K3698 (∆*oprM*) ± TXA09155 ^#^
Cefpirome	4	8	8	2
Levofloxacin	8	16	16	2
Cotrimoxazole	8	32	16	4
Doxycycline	>8	≥64	32	8
Minocycline	>4	>128	64	16
Chloramphenicol	4	32	16	8
Imipenem *	1	1	1	1

^#^ TXA09155 concentration = 6.25 µg/mL; * Not a substrate of RND efflux pumps in *P. aeruginosa.*

**Table 4 antibiotics-11-00716-t004:** Frequency of resistance to TXA09155 and levofloxacin.

Strain	Levofloxacin (4 μg/mL)	Levofloxacin (4 μg/mL) + TXA01182 (12.5 μg/mL)
*P. aeruginosa* ATCC 27853	4.97 × 10^−7^	<3.33 × 10^−9^

**Table 5 antibiotics-11-00716-t005:** Genetic study of TXA09155 resistance in *P. aeruginosa*.

Resistance to:	Parent Strain	FoR	Strain Name	Mutation	Gene Role
TXA09155 alone (4×-MIC)	*P. aeruginosa* ATCC 27853	1.40 × 10^−6^	EPIR1S	*phoQ*-L175R	Two-component regulatory system
			EPIR9S	*phoQ*-L175R	
			EPIR20L	*phoQ*-H248D	
TXA09155 alone (1×-MIC)		1.90 × 10^−3^	EPIR43	*ompH*-Q127X *	Skp-like periplasmic chaperone
TXA09155 (⅛×-MIC) + LVX (1×-MIC)	*P. aeruginosa* DA7232	2.48 × 10^−8^	EPIR24L	*trpS*-R171S	Tryptophan-tRNA ligase

FoR, frequency of resistance; * Termination codon.

**Table 6 antibiotics-11-00716-t006:** ADMET properties of TXA09155.

MW	Solubility at pH 7.4 (μM)	cLogP	HLM and RLM Metabolic Stability	CYP Inhibition IC_50_ (μM)				
				CYP 1A2	CYP 2C19	CYP 2C9	CYP 2D6	CYP 3A4
366	155 ± 3	2.6	>60%	>100	>100	>100	67.2	28.5

MW, molecular weight; HLM, human liver microsomes; RLM, rat liver microsomes.

## Data Availability

Data are contained within the article or Appendix A. The data presented in this study are available online at Appendix A.

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
