# Peer review of "Evaluation of a Conformationally Constrained Indole Carboxamide as a Potential Efflux Pump Inhibitor in Pseudomonas aeruginosa"

_antibiotics, 2022, doi:10.3390/antibiotics11060716_

Round 1

Reviewer 1 Report

The original article entitled “Evaluation of a Conformationally Constrained Indole Carboxamide as Potential Efflux Pump Inhibitor in Pseudomonas aeruginosa” is focused on the characterization of the antibacterial and antibiotic-potentiating activity of new efflux pump inhibitors against P. aeruginosa. I believe that the manuscript is of high quality, well described and contains many different research methods confirming the hypothesis presented by the Authors. T further improve the quality of the manuscript, below I would like to present a short list of suggestions:

  • Right at the beginning, as the first paragraph, the Authors should describe Pseudomonas aeruginosa in terms of clinical relevance (diseases' development, virulence) and the easiness of acquiring resistance. There is no such description in the introduction, while this bacterium was used as a model in the present manuscript.
  • Table 1: I recommend modifying the table to change: 1. "No EPI" to "Lack of EPI" or "EPI-negative"; 2. Change "+ TXA01182 (6.25 µg / mL)" to "+ 6.25 µg / mL of TXA01182 (fold change)"; 3. Change "+TXA09155 (6.25 µg/mL)" to "+ 6.25 µg / mL of TXA09155 (fold change)"; 4. In the lines with data containing the MIC values, please add in parentheses the fold reduction of the MIC relative to the control
  • Table 2: I propose to remove the "TXA09155 MIC (μg/ml) 50" column and modify the title to include this information
  • I believe that it is definitely worth extending the article with statistical calculations. This would greatly assist in the interpretation of the data presented in Figures 2-6

Author Response

The original article entitled “Evaluation of a Conformationally Constrained Indole Carboxamide as Potential Efflux Pump Inhibitor in Pseudomonas aeruginosa” is focused on the characterization of the antibacterial and antibiotic-potentiating activity of new efflux pump inhibitors against P. aeruginosa. I believe that the manuscript is of high quality, well described and contains many different research methods confirming the hypothesis presented by the Authors. T further improve the quality of the manuscript, below I would like to present a short list of suggestions:

  • Right at the beginning, as the first paragraph, the Authors should describe Pseudomonas aeruginosa in terms of clinical relevance (diseases' development, virulence) and the easiness of acquiring resistance. There is no such description in the introduction, while this bacterium was used as a model in the present manuscript. Done
  • Table 1: I recommend modifying the table to change: 1. "No EPI" to "Lack of EPI" or "EPI-negative"; 2. Change "+ TXA01182 (6.25 µg / mL)" to "+ 6.25 µg / mL of TXA01182 (fold change)"; 3. Change "+TXA09155 (6.25 µg/mL)" to "+ 6.25 µg / mL of TXA09155 (fold change)"; 4. In the lines with data containing the MIC values, please add in parentheses the fold reduction of the MIC relative to the control Done
  • Table 2: I propose to remove the "TXA09155 MIC (μg/ml) 50" column and modify the title to include this information Done
  • I believe that it is definitely worth extending the article with statistical calculations. This would greatly assist in the interpretation of the data presented in Figures 2-6 Modified legends

Reviewer 2 Report

The manuscript "Evaluation of a Conformationally Constrained Indole Carboxamide as Potential Efflux Pump Inhibitor in Pseudomonas aeruginosa" is a well written study that intends to determine the suitability of TXA09155, a novel indole derivative, to be used as a novel efflux pump inhibitor against Pseudomonas aeruginosa. The study has a logical and elegant experimental design, reports novel interesting advances, has a good presentation and it is written in a good English. Therefore, it can be published in Antibiotics after fixing the following concerns:

1) Lines 78-84. Some references could be provided related to the effect of the conformation constrainst and molecular flexibility on the activity of flexible molecules.

2) Figure 5. Please indicate the subfigure 5c in the legend. Looks like that it is described, but the "(C)" caption is missed in the legend.

3) Line 364, please correct the typo "bodyand".

4) Indicate the details of equipment used as NMR or LC-MS.

5) Indicate, in a general form (not necessary to be detailed for each reagent), the providers of the chemicals used in the synthesis.

6) Please revise that all providers (of equipments / reagents / biological kits and so on) are cited in the same format: name of company (city, country).

7) LC-MS experiment has been performed. Authors should provide the purity of the final compound, calculated through the integration of the peak are in chromatogram.

Author Response

The manuscript "Evaluation of a Conformationally Constrained Indole Carboxamide as Potential Efflux Pump Inhibitor in Pseudomonas aeruginosa" is a well written study that intends to determine the suitability of TXA09155, a novel indole derivative, to be used as a novel efflux pump inhibitor against Pseudomonas aeruginosa. The study has a logical and elegant experimental design, reports novel interesting advances, has a good presentation and it is written in a good English. Therefore, it can be published in Antibiotics after fixing the following concerns:

1) Lines 78-84. Some references could be provided related to the effect of the conformation constrainst and molecular flexibility on the activity of flexible molecules. Done

2) Figure 5. Please indicate the subfigure 5c in the legend. Looks like that it is described, but the "(C)" caption is missed in the legend. Done

3) Line 364, please correct the typo "bodyand". Done

4) Indicate the details of equipment used as NMR or LC-MS. Done

5) Indicate, in a general form (not necessary to be detailed for each reagent), the providers of the chemicals used in the synthesis. Done

6) Please revise that all providers (of equipments / reagents / biological kits and so on) are cited in the same format: name of company (city, country). Done

7) LC-MS experiment has been performed. Authors should provide the purity of the final compound, calculated through the integration of the peak are in chromatogram. Done

Reviewer 3 Report

Dears authors 

Approaches to combat efflux-mediated multidrug resistance involve, in part, the development of indirect antimicrobial agents capable of inhibiting efflux thereby saving the activity of antimicrobials previously inactivated by efflux. Here we present TXA09155, a novel efflux pump inhibitor (EPI) formed by conformationally binding our previously reported EPI TXA01182. TXA09155 demonstrates strong potentiation in combination with multiple antibiotics with efflux responsibility against wild-type and multidrug-resistant P. aeruginosa (MDR). At 6.25 µg / mL, TXA09155 showed ≥8-fold potentiation of levofloxacin, moxifloxacin, doxycycline, minocycline, cepyroma, chloramphenicol and co-trimoxazole.

 Introduction: it must be reformed in the content and in the writing of the general part to review the syntax of the topic . Discussion: to deepen in consideration of the problem of antibiotic resistance and the correlation between the ability to form biofilm, virulence factors and phonotypic resistance, use to deepen these studies: Learn more about this by using and citing the following references: PMID: 34572716 ; PMID: 35321081; PMID: 35453262  . Check the bibliographic entries throughout the text, some of which are non-compliant, review some entries in the bibliographic references  Review the English grammar and in particular the applied scientific English: in particular, the verb tenses and the syntax in the discussion.

Author Response

Approaches to combat efflux-mediated multidrug resistance involve, in part, the development of indirect antimicrobial agents capable of inhibiting efflux thereby saving the activity of antimicrobials previously inactivated by efflux. Here we present TXA09155, a novel efflux pump inhibitor (EPI) formed by conformationally binding our previously reported EPI TXA01182. TXA09155 demonstrates strong potentiation in combination with multiple antibiotics with efflux responsibility against wild-type and multidrug-resistant P. aeruginosa (MDR). At 6.25 µg / mL, TXA09155 showed ≥8-fold potentiation of levofloxacin, moxifloxacin, doxycycline, minocycline, cepyroma, chloramphenicol and co-trimoxazole.

Introduction: it must be reformed in the content and in the writing of the general part to review the syntax of the topic . Will the reviewer please be more specific about which sentences s(he) refers to?

Discussion: to deepen in consideration of the problem of antibiotic resistance and the correlation between the ability to form biofilm, virulence factors and phonotypic resistance, use to deepen these studies: Learn more about this by using and citing the following references: PMID: 34572716; PMID: 3532108; PMID: 35453262. Done

Check the bibliographic entries throughout the text, some of which are non-compliant, review some entries in the bibliographic references Done

Review the English grammar and in particular the applied scientific English: in particular, the verb tenses and the syntax in the discussion. Will the reviewer please be more specific about which sentences s(he) refers to?

Round 2

Reviewer 3 Report

Been made the corrections